# Study of Catalytic CO_2_ Absorption and Desorption with Tertiary Amine DEEA and 1DMA-2P with the Aid of Solid Acid and Solid Alkaline Chemicals

**DOI:** 10.3390/molecules24061009

**Published:** 2019-03-13

**Authors:** Huancong Shi, Min Huang, Qiming Wu, Linna Zheng, Lifeng Cui, Shuping Zhang, Paitoon Tontiwachwuthikul

**Affiliations:** 1Department of Environmental Science and Engineering, College of Science, University of Shanghai for Science and Technology, Shanghai 200093, China; 172681815@st.usst.edu.cn (M.H.); 19921260976@163.com (Q.W.); linna1234576@163.com (L.Z.); 2Clean Energy Technology Research Innovation (CETRI), Faculty of Engineering and Applied Science, University of Regina, 3737 Wascana Parkway, Regina, SK S4S 0A2, Canada

**Keywords:** CO_2_ absorption, tertiary amines, CO_2_ desorption, energy cost, equilibrium, solubility

## Abstract

Studies of catalytic CO_2_ absorption and desorption were completed in two well-performed tertiary amines: diethylmonoethanolamine (DEEA) and 1-dimethylamino-2-propanol (1DMA-2P), with the aid of CaCO_3_ and MgCO_3_ in the absorption process, and with the aid of γ-Al_2_O_3_ and H-ZSM-5 in the desorption process. The batch process was used for CO_2_ absorption with solid alkalis, and the recirculation process was used for CO_2_ desorption with solid acid catalysts. The CO_2_ equilibrium solubility and pKa were also measured at 293 K with results comparable to the literature. The catalytic tests discovered that the heterogeneous catalysis of tertiary amines on both absorption and desorption sides were quite different from monoethanolamine (MEA) and diethanolamine (DEA). These results were illustrative as a start-up to further study of the kinetics of heterogeneous catalysis of CO_2_ to tertiary amines based on their special reaction schemes and base-catalyzed hydration mechanism.

## 1. Introduction

The global warming and sudden change of climates have driven scientists and engineers to develop cost-effective processes for CO_2_ removal from coal-fired power plants [1]. The chemical absorption of CO_2_ in the post-combustion CO_2_ capture process may be implemented on the commercial scale [2]. This absorption process enables the CO_2_ removal with “energy efficient” amine solvents via an absorption-desorption unit [1]. 

The development of attractive and novel amines has been a strong drive to meet these basic requirements: high absorption rates, large cyclic capacity, and low regeneration energy [3,4]. Amine solutions of monoethanolamine (MEA), diethanolamine (DEA) and methyldiethanolamine (MDEA) have been widely used for CO_2_ removal, as benchmarks of primary, secondary, and tertiary amines [5]. MEA exhibits a higher reaction rate, but smaller cyclic capacity, higher energy costs for regeneration and higher corrosion rates [4]. To overcome these limitations, MEA is usually blended with a variety of tertiary amines in preparation for improved solvents called “MEA-R_3_N blends” such as MEA-MDEA [6,7], MEA-4-diethylamino-2butabol (DEAB) [6,7], MEA-diethylmonoethanolamine (DEEA) [8] and MEA-1-dimethylamino-2-propanol (1DMA-2P), etc. [9]. Among these MEA-R_3_N blends, the concentration of MEA is usually 5.0 mol/L, but the concentration of tertiary amine is around 1.0–2.0 mol/L for MDEA [6,7], and 1.0–1.5 for amines such as DEEA (1.0 mol/L) [9], 1DMA-2P (1.0 mol/L) [9] of DEAB (1.25 mol/L) [6,7] etc. In industry, the tertiary amines are usually prepared at a concentration of 1–1.5 mol/L for absorption if blended with a concentrated amine such as 5.0 mol/L MEA.

Most tertiary amines fulfill two basic requirements: a large cyclic capacity with a relatively lower regeneration energy compared with MEA and DEA [5]. Among these commercial tertiary amines, MDEA is always used as benchmark. Recently, two good performance tertiary amines have drawn research interest: DEEA and 1DMA-2P [1,5]. DEEA is studied for its relatively higher CO_2_ equilibrium solubility than MDEA [10]. The pKa of it [11] is studied, and its mass transfer performance [12] is also better than MDEA. 1DMA-2P is also investigated for its large CO_2_ cyclic capacity, much higher absorption rates (kinetics) [13] and better mass transfer characteristics [14] than MDEA. It also has a much lower CO_2_ absorption heat than MEA, DEA, PZ and MDEA [15]. Moreover, the equilibrium solubility, the pKa [16] and ion speciation plots of 1DMA-2P-CO_2_-H_2_O are generated with ^13^C NMR methods [17].

Meanwhile, based on recent studies, the effects of solid base chemicals to MEA (K/MgO, CaCO_3_ and MgCO_3_) [18,19] and DEA (CaCO_3_ and MgCO_3_) [20] on CO_2_ absorption have been verified. They accelerate the CO_2_ absorption rates. However, the effects of both solid bases onto a tertiary amine have rarely been discussed either. The effects of solid acid catalysts (γ-Al_2_O_3_, H-ZSM-5, TiO(OH)_2_, and transition metal oxides V_2_O_5_, MoO_3_, WO_3_, TiO_2_, and Cr_2_O_3_) to MEA [9,21,22,23,24,25] and DEA [26] have also been studied and proven to be effective in the reduction of heat duty. Some studies have been completed to investigate the effects of solid acid catalysts (H-ZSM-5, MCM-41 and SO_4_^2−^/ZrO_2_) with blended amine solvents of 5 + 1.0 mol/L MEA-DEEA and MEA-1DMA-2P [9]. However, the effects of the heterogeneous catalysis toward CO_2_-R_3_N alone require detailed investigation.

The catalytic effects were verified of CaCO_3_ and MgCO_3_ toward MEA (19) and DEA (20) on CO_2_ absorption, and also (γ-Al_2_O_3_ and H-ZSM-5) to MEA [9,21,22,23] and DEA [26] on CO_2_ desorption. These catalytic effects toward tertiary amines were the purpose of this study. Since the reaction schemes and the mechanism of CO_2_ absorption with tertiary amines are quite different from primary amine (MEA) or secondary amine (DEA), these differences make the effects of heterogeneous catalysis worthy of deep investigation with solid bases and solid acids as a start-up. For this study, the CO_2_ absorption with 1.0–1.5 mol/L DEEA and 1DMA-2P solvents were investigated with the aid of a solid base (CaCO_3_ and MgCO_3_), and CO_2_ desorption 1.0–2.0 mol/L DEEA and 1DMA-2P solvents for solid acids (γ-Al_2_O_3_ and H-ZSM-5). The effects of heterogeneous catalysis on both sides of absorption and desorption were studied and compared with MEA and DEA.

## 2. Theory

### 2.1. Reaction Scheme, and Suitable Mechanisms of CO_2_-R_3_N Interaction

The reaction scheme of the CO_2_ reaction with tertiary amine (R_1_R_2_R_3_N) is presented below with Equations (1)–(6) [3]. Equation (1) is the major reaction being emphasized. Different from primary/secondary amines (R_1_NH_2_/R_1_R_2_NH), the major anion is bicarbonate (HCO_3_^−^) instead of carbamate (R_1_R_2_N-COO^−^).
CO_2_ + R_1_R_2_R_3_N + H_2_O ↔ R_1_R_2_R_3_NH^+^ + HCO_3_^−^(1)
R_1_R_2_R_3_NH^+^ ↔ R_1_R_2_R_3_N + H^+^(2)
CO_2_ + H_2_O ↔ H_2_CO_3_ ↔ H^+^ + HCO_3_^−^(3)
CO_2_ + OH^−^ ↔ HCO_3_^−^(4)
HCO_3_^−^ ↔ H^+^ + CO_3_^2−^(5)
H_2_O ↔ H^+^ + OH^−^(6)

Based on a recent review of kinetics [27], the Zwitterion mechanism [28] and Termolecular mechanism [29] are suitable for CO_2_ reactions with primary and secondary amines. The mechanism of CO_2_ reaction with tertiary amines was proposed by Donaldson and Nguyen and is termed the “base-catalyzed hydration mechanism” [30]. The tertiary amine (R_1_R_2_R_3_N) does not react directly with CO_2_, but rather acts as a base that catalyzes the hydration of CO_2_ [30] based on Equation (1).

The rate equation was written as Equation (7), with the rate constant of tertiary amine (R_3_N) of $k_{R_{3}N}$: [31] from Equation (7), the bigger amine concentration results in higher absorption rates.
(7)$$r_{{\mathrm{CO}}_{2}}=k_{R_{3}N}\left[R_{3}N\right]\left[{\mathrm{CO}}_{2}\right].$$

Moreover, there are three balance equations in the R_3_N-CO_2_-H_2_O systems, as presented in Equations (8)–(10) [3].

Mass balance of R_3_N:
[R_1_R_2_R_3_N]_0_ = [R_1_R_2_R_3_NH^+^] + [R_1_R_2_R_3_N](8)

Mass balance of Carbon, where α is CO_2_ loading:
α × [R_1_R_2_R_3_N]_0_ = [CO_2(aq)_] + [HCO_3_^−^] + [CO_3_^2−^](9)

Charge balance:
[R_1_R_2_R_3_NH^+^] + [H^+^] = [OH^−^] + [HCO_3_^−^] + 2[CO_3_^2−^](10)

### 2.2. Role of Solid Alkalis Chemicals for Absorption

The catalytic effects of both solid alkalis to MEA and DEA were already studied [19,20]. However, the role was slightly different with tertiary amines. Since the reaction mechanism of CO_2_-R_3_N was “base-catalyzed hydration” [30], reactions (1)–(6) could be facilitated with either a liquid base [OH^−^] or solid alkalis/Lewis base [19] due to the acidic chemical nature of CO_2_. Therefore, the solid alkalis (CaCO_3_ and MgCO_3_) could enhance the hydration of CO_2_, and proton transfer of [H_2_CO_3_] to water or R_3_N, as an aid to the liquid base of [OH^−^].
(11)$${\mathrm{CO}}_{2}+H_{2}O\overset{\mathrm{Base}}{↔}H^{+}+{{\mathrm{HCO}}_{3}}^{-}$$
(12)$${\mathrm{CO}}_{2}+R_{1}R_{2}R_{3}N+H_{2}O\overset{\mathrm{Base}}{↔}H^{+}+{{\mathrm{HCO}}_{3}}^{-}$$

After detailed investigations, the role of solid alkalis was “Lewis base” and “proton acceptor” [19] to facilitate proton transfer of H_2_CO_3_, with reactions below:
H_2_CO_3_ + Catalyst ↔ HCO_3_^−^ + Cat-[H^+^](13)
Cat-[H^+^] + R_3_N ↔ Catalyst + [R_3_NH^+^](14)

The solid catalyst (CaCO_3_/MgCO_3_) accepted the protons [H^+^] from H_2_CO_3_ with its long pair of electrons on the “O atom” via (13). After that, the proton was transferred from the solid catalyst to the tertiary amine (R_3_N) via (14), because the tertiary amine is a stronger base than Lewis base (CaCO_3_/MgCO_3_). The overall reaction was (3) + (13) + (14) = (1), and the solid catalyst was involved in the reaction but did not change the reactant or products. From Equations (13) and (14), with an increased amount of solid base, the reaction rate of (13) would increase but the rate of (14) would decrease. With an increased mass of catalysts, the protons [H^+^] are easier to transfer onto a solid surface, but harder to release back to R_3_N. Therefore, there is an optimized amount of solid catalyst for CO_2_ absorption, after that, the rates slightly decreased.

### 2.3. Role of Lewis Acid and BrØnsted Acid for CO_2_ Desorption

The role of Lewis acid (γ-Al_2_O_3_) and BrØnsted acid for CO_2_ desorption with MEA and DEA has already been discussed repeatedly [9,21,26].

However, the role of both acids needs to be discussed for CO_2_ desorption with tertiary amines because the reactions were different, and no carbamate was involved. After the study of reaction schemes, the effect and mechanism of both solid acids were discussed as follows:

γ-Al_2_O_3_ as catalyst:
Al_2_O_3_ + 2OH^−^ ↔ 2 AlO_2_^−^ + H_2_O(15)
R_1_R_2_R_3_NH^+^ + AlO_2_^−^ ↔ R_1_R_2_R_3_N + HAlO_2_(16)
HAlO_2_ + HCO_3_^−^ ↔ AlO_2_^−^ + H_2_O + CO_2_ ↑(17)

H-ZSM-5 as catalyst:
H-ZSM-5 + HCO_3_^−^ ↔ (ZSM-5)^−^ + H_2_O + CO_2_ ↑(18)
R_1_R_2_R_3_NH^+^ + (ZSM-5)^−^ ↔ R_1_R_2_R_3_N + H-ZSM-5(19)

Both solid acids could facilitate the CO_2_ production rates and reduce the energy costs. In short, the Al_2_O_3_ is Amphoteric Oxide, and it was converted into AlO_2_^−^ in basic solutions. From the published energy diagrams [6], Al_2_O_3_ (AlO_2_^−^) can speed up the proton transfer from R_1_R_2_R_3_NH^+^ to HCO_3_^−^ and CO_2_ generation under heat. H-ZSM-5 is the proton donor, which directly provides protons to the solvent and facilitates CO_2_ generation.

Based on the reaction schemes above, both solid acids involve two steps, namely “accept proton” and “donate/transfer proton”. Since γ-Al_2_O_3_ contains no proton, it has to “accept proton” first and “transfer proton” later in the desorption process. The H-ZSM-5 contains proton itself, and it intends to “donate proton” first and then “accept proton” from R_1_R_2_R_3_NH^+^. The mechanisms are similar for both solid catalysts but the order of “accept proton” and “donate/transfer proton” is opposite.

## 3. Materials and Experimental Methods

### 3.1. Chemicals

The solid chemicals were purchased from Huishan Chemical Ltd; they were CaCO_3_ and MgCO_3_. The CO_2_ gas and the liquid chemicals DEEA and 1-DMA2P were purchased from Tansoole Chemical Ltd. (Shanghai, China). HCl and methyl orange were commercially obtained from Guoyao Chemical Ltd. (Shanghai, China). The chemical structures and full name of DEEA and 1DMA2P were presented elsewhere [5].

### 3.2. pKa Analysis

The titration technique was adopted to determine the amine dissociation constant (Ka) with standard 1 mol/L HCl [32,33,34,35]. This is a simplified pH method to test the pKa of different amines under different temperatures. For an aqueous amine solution, the Equation (20) below showed the deprotonation reaction of AmineH^+^/Amine as a conjugated pair of acid-bases.
(20)$${\mathrm{AmineH}}^{+}\overset{k_{a}}{↔}\mathrm{Amine}+H^{+}$$

Based on a detailed pKa study of tertiary amines [11] the pH meter measured the activity {R_3_NH^+^} of amine solvents instead of its real concentration [R_3_NH^+^]. The correlation is {R_3_NH^+^} = [R_3_NH^+^] $\gamma_{{\mathrm{BH}}^{+}}$ [11]. This study assumed that this diluted amine solvent (<0.10 mol/L) is the ideal solution (when the concentration is very low and the activity coefficient $\gamma_{{\mathrm{BH}}^{+}}$ equals to 1 {BH^+^} ≈ [BH^+^]) [11]. Then, Equations (21) and (22) below were used to calculate the Ka.
(21)$$K_{a,\mathrm{Amine}}=\frac{\left[\mathrm{Amine}\right]\left[H^{+}\right]}{\left[{\mathrm{AmineH}}^{+}\right]}$$
(22)$$pK_{a}=-\log(K_{a,\mathrm{Amine}})=-\log(\left[\mathrm{Amine}\right]\left[H^{+}\right]/\left[{\mathrm{AmineH}}^{+}\right])$$

The pH meter was used to measure the concentration of H^+^ in the solution [32,33,34,35]. As presented in Equation (23), the disappearance of H^+^ during the titration process resulted from its reaction with Amine to generate AmineH^+^, and reaction (20) is the dominant in aqueous solution. A mass balance of protons as shown in Equation (23) was adopted to calculate the concentration of AmineH^+^, and the amine balance equation as shown in Equation (24) was adopted to calculate free amine.
(23)$$\mathrm{nHCl}-\left[H^{+}\right]V_{\mathrm{total}}=\left[{\mathrm{AmineH}}^{+}\right]V_{\mathrm{total}}$$
(24)$$(\left[\mathrm{Amine}\right]+\left[{\mathrm{AmineH}}^{+}\right])V_{\mathrm{total}}=n_{0,\mathrm{Amine}}$$

In Equations (23) and (24) above, nHCl is the number of moles of HCl added during the titration process, V_total_ is the total liquid volume after the titration process, and n_0,Amine_ is the moles of free amines as a start, which can be determined by titration with 1.0 mol/L HCl until the indicator of methyl orange turns pink.

For the experiment, the Ka of amine was determined based on the procedure described [32]. Briefly speaking, 100 mL of 0.10 mol/L amine solution was carefully prepared and titrated with 10 mL of 1.0 mol/L HCl standard solution at 298 K until the end point was observed. During the titration process, the pH meter was placed in the solution to record pH value with the addition of 1 mL HCl each time. A table of pH value and amount of HCl was generated. Equations (21)–(24), were used to determine the concentration of [AmineH^+^], [Amine] and then calculate the dissociation constant (Ka). The dataset was only recorded at pH > 9, because the results would be inaccurate if pH < 9, where the generation of [H^+^] or [OH^−^] from water is not negligible.

### 3.3. CO_2_ Absorption Process with Absorption Profiles

A set of stirred-cell reactors were built in the lab, with the flow chart exhibited in Appendix A. The process was similar to that of other studies, [17,20] and the diameter of the reactor is 11.0 cm (a constant interfacial area of 95.0 cm^2^). The solid alkalis (CaCO_3_ and MgCO_3_) were pelletized at 2–3 mm and wrapped into small balls with a diameter of 2.5 cm, each ball contained about 2.5 g, similarly [20].

The CO_2_ absorption process was similar to that of other works [17,20]. Three-hundred milliliters of amine solvent was prepared at a concentration of 1.0–1.5 mol/L. For 1DMA-2P, it started to crystalize at 2.0 mol/L at 293 K, then 2.0 mol/L was not tested for absorption. The CO_2_ gas flow was introduced into the reactor at a rate of 1.5 L/min. The PCO_2_ was 101.3kPa with 100% purity. The operation temperature was maintained at 20 °C by the cooled water bath. The process was connected to the air with the pressure of 1 atm. Some vials were prepared and 2 mL samples were pipetted every 3–5 min into each of them. The titration technique was adopted to test the CO_2_ loadings and the results were recorded within 3−5 min [17]. A Chittick apparatus with an AAD of 2.5% was adopted to conduct the CO_2_-loading tests of the samples [36]. In order to ensure repeatability, these tests were performed at least twice. 

We already verified the catalytic effect of CaCO_3_ with CO_2_ absorption with 1.0 mol/L MEA in another study [19]. The results exhibited that the order of catalytic effects was: 5 g CaCO_3_ in gas-liquid interface >5 g, CaCO_3_ in bulk of liquid >0 g, CaCO_3_ ≈ intert stainless steel. From the BET tests of CaCO_3_ and MgCO_3_, the surface areas were 0.428 m^2^/g for CaCO_3_ and 9.498 m^2^/g for MgCO_3_. The pore diameters were 31.3 nm and 4.31 nm, which facilitated the external mass transfer of amine molecules onto a solid surface. The inert material with large surface area might have a significant effects of mass transfer causing the increase of CO_2_ absorption. However, it could not replace the role of “Lewis base“ or “proton acceptor“ as solid alkaline catalysts to enhance the CO_2_ absorption with tertiary amine. 

After the absorption profiles were plotted with (α, time), the initial absorption rates (I_abs_rate_, mol CO_2_/min) [34] are determined as the slope of the linear regression of absorption profiles was at data range of CO_2_ loadings of 0.0–0.20 mol/mol, in Equation (25):
(25)$$I_{\mathrm{abs}\_\mathrm{rate}}=C\times V\frac{d\alpha }{dt}$$
where “C” is the Concentration and “V” is the Volume, and α is the CO_2_ loading. 

The initial absorption rates were adopted in this study to compare the CO_2_ absorption performance of different cases of catalysts. These data were generated at a consistent level for different catalytic and non-catalytic cases. The results were inadequate for kinetic studies for now, but it was adequate to verify the catalytic effect as a start up. 

### 3.4. CO_2_ Desorption Tests with Heat Duty Calculation

An open recirculation-process (Appendix A) vessel equipped with an electrometer [9,26,37] was adopted to conduct the CO_2_ desorption tests under atmosphere to extract DEEA and 1DMA-2P solvents at 1.0 mol/L, 1.5 mol/L and 2.0 mol/L, and two types of solid acid catalyst were used as γ-Al_2_O_3_ and H-ZSM-5 representing Lewis acids and Brønsted acids [9,26]. The acidic catalytic conditions were 5.0 g, 7.5 g and 10.0 g, This CO_2_ desorption process was similar to that of others in the literature [9,26]. In this study, 250 mL of the amine solvent was put into the flask with a volume of 500 mL. The CO_2_ loading was over 0.80 mol/mol in preparation for desorption, with CO_2_ introduced into amine solvents beforehand. Small balls of various catalysts were placed into the solvents as well. 

The experimental procedures were similar to those in our previous study [26]. The process was stirred and heated to 363 K. Based on the analysis of the CO_2_ loading of samples at 0−4 h, the catalytic effects on CO_2_ desorption were evaluated. A vial into which the samples were pipetted was then cooled down in a cooled water bath so as to maintain the CO_2_ loading. Similar to absorption, the CO_2_ loading was tested immediately after sample collection by titration [26]. A Chittick apparatus [36] was adopted to conduct the CO_2_-loading tests of each sample and they were performed at least twice so as to ensure repeatability. The average CO_2_ loading was then plotted for each run. 

As part of the pivotal desorption parameter [9,26,33], the heat duties were calculated from CO_2_ production with Equations (26) and (27) below [9,21,26]. The nCO_2_ (mol) is the amount of desorbed CO_2_, α (mol of CO_2_ per mole of amine) is the CO_2_ loading, and C (mol/L) and V (L) are the concentration and volume of amine solvent. This method of calculating heat duties was similar to that in other studies [9,21,26].
(26)$$H=\frac{\mathrm{heat}\;\mathrm{input}/\mathrm{time}}{{\mathrm{amount}\;\mathrm{of}\;\mathrm{CO}}_{2}/\mathrm{time}}=\frac{E}{{\mathrm{nCO}}_{2}}\;\frac{\mathrm{kJ}\;\mathrm{of}\;\mathrm{electricity}/h}{{\mathrm{mol}\;\mathrm{of}\;\mathrm{CO}}_{2}/h}$$
(27)$${\mathrm{nCO}}_{2}=\left(\alpha_{\mathrm{rich}}-\alpha_{\mathrm{lean}}\right)C\times V$$

## 4. Results and Discussions

### 4.1. The Critical Point of CO_2_ Absorption Curve of DEEA and 1-DMA-2P at 293 K, Affected by Equilibrium Solubility

The CO_2_ equilibrium solubility at different temperatures and pressures was one of the key parameters for screen solvent [15]. Although different solid base chemicals could accelerate the CO_2_ absorption rates, they could hardly shift the CO_2_ equilibrium solubility under the VLE model, which was only determined by temperature [10]. 

There are two common methods to generate the VLE model of tertiary amines, (1) the predictive model by simulation using MDEA as benchmark [38,39], (2) experimental studies of CO_2_ solubility with absorption [5,10,15]. The equilibrium solubility of DEEA and 1DMA-2P was reported based on long-term vapor liquid equilibrium experiments at 298–313 K [5]. The accurate equilibrium solubility was relatively high due to the experimental procedures. Luo et al. completed the experiments and the modeling of data of DEEA-CO_2_-H_2_O at 1.0 to 4.0 mol/L under vapor-liquid equilibrium, from 293–353 K [10]. They used 0.15 L/min mixed gas of CO_2_/N_2_, with CO_2_ partial pressures from 6.2 kPa to 100.8 kPa and maintained 8–10 h to reach the thermodynamic equilibrium conditions. By the long-term tests, the equilibrium solubility of DEEA was 0.971 mol/mol at 293 K with P_CO2_ of 100.8 kPa. Similarly, Liu et al. completed the modeling of CO_2_ equilibrium solubility of 1DMA-2P solution, with CO_2_ partial pressures from 8 kPa to 101.3 kPa at 1, 2, and 5 mol/L, 298–333 K [15]. For 1.0 mol/L 1DMA-2P, the solubility was reported as 1.02 mol/mol at 101.3 kPa at 298 K with 8 h operation [15]. 

In this study, we determined the “critical point of CO_2_ absorption curves” based on the slope of CO_2_ absorption curves of 1.0 mol/L amine without catalysts. It was affected by the CO_2_ solubility or “Ion speciation plot” of R_3_N-CO_2_-H_2_O systems under the Vapor-liquid equilibrium model. From the CO_2_ absorption curves, different stages of the reaction were contained, (1) CO_2_ + R_3_N + H_2_O around 0–0.85 mol/mol, and (2) CO_2_ + H_2_O above 0.85 mol/mol when free R_3_N was exhausted. The slope of absorption curves turned very flat after this critical point, indicating that all the amines were converted to amineH^+^, and CO_2_ only reacted with water afterward. The critical point was determined by the graphic method based on the cross of slopes at different stages (Appendix A). 

Therefore, the “critical point of CO_2_ absorption curves” was calculated as about 0.87 mol/mol for DEEA, and 0.81 mol/mol for 1DMA-2P at 1.0 mol/L and 293 K here was based on the graphic method. The CO_2_ solubility of DEEA and 1DMA-2P under Vapor-Liquid Equilibrium (VLE) model was plotted in Figure 1 at 298–313 K [5]. Our data were added in Figure 1, but these data were not “CO_2_ equilibrium solubility”. It was affected by the “CO_2_ equilibrium solubility” and “Ion speciation plots”. From the literature value, CO_2_ equilibrium solubility of CO_2_ is 0.839 mol/mol for DEEA, and 0.789 mol/mol for 1DMA-2P at 298 K [5]. The trend was consistent from Figure 1 at 298–313 K: with an increase of temperature, the solubility of CO_2_ was slightly decreasing [5,10].

### 4.2. The pKa of DEEA and 1-DMA-2P at 293 K

The pKa is also an important parameter for tertiary amines, which can be used for the selection of amine solutions for both CO_2_ removal and the study of the reaction kinetic mechanism [15]. Based on the base-catalyzed mechanism, tertiary amines (R_3_N) do not directly react with CO_2_, but absorbed protons from H_2_CO_3_. The simplified pH method for the detection of pKa is quite similar to that of other studies [32,33,34,35]. It excluded the data of pH < 9.0. This was because the conjugated acid/base of [Amine]/[AmineH^+^] did not exist under acidic conditions (pH < 7.0). Moreover, a pH value between 7–9 was not selected either, for the calculation of pKa in Equation (22) was based on the assumption that the [H^+^] released into the solution was 100% from [AmineH^+^] with neglect of proton release from H_2_O. Meanwhile, the [OH^−^] in the water solution was mainly from proton transfer from H_2_O to Amine, and [OH^−^] released from H_2_O was also negligible. In the case of pH < 9, the [OH^−^] in the solution was smaller than 10^−5^ mol/L, which was not 100 times bigger than the [OH^−^] (10^−7^ mol/L) dissociated by neutral H_2_O. Then the dissociation of [OH^−^] from H_2_O was not negligible, and thus Equation (22) had errors. Finally, the pKa was measured and grouped into Table 1 and Figure 2 for comparison. 

Hence, the pKa was measured as 9.82 for DEEA at 293 K. This value was the same as the literature value of 9.82 at 293 K, [11] which reflected the accuracy. The pKa was measured as 9.51 for 1DMA-2P at 293 K, comparable to 9.41 at 298 K [15] from K_2_ correlation model for solubility study. It was measured as 9.67 at 301 K based on ^13^C NMR analysis. [17] Different methods might result in slight deviations. Recently, Liu et al developed a linear calibration of pKa of 1DMA-2P in Equation (27) at 298–333 K [15]. We combined these data with our own results and plotted in Figure 2. Our data (9.51, 293 K) was outside the range of that calibration curve, but the data was consistent with the line. The new linear calibration was generated in Equation (28) to expand the pKa of 1DMA-2P at 293-333 K.
(28)$$\mathrm{pKa}=\frac{2639}{T}+0.559;\;298-333\;K$$
(29)$$\mathrm{pKa}=\frac{2545}{T}+0.850;\;293-333\;K$$

### 4.3. The CO_2_ Absorption Profiles with Initial Absorption Rates

The CO_2_ absorption profiles of 1.0 mol/L and 1.5 mol/L DEEA and 1DMA-2P solvents were plotted in Figure 3, Figure 4, Figure 5 and Figure 6, with the aid of CaCO_3_ and MgCO_3_, respectively. The optimized mass of solid alkalis under various amine concentrations was presented in Table 2. The optimized mass was based on the catalytic reactions (13) and (14), with explanation in Section 2.2. With an increased mass of solid catalysts, the initial absorption rates increased first, reached an optimum and then decreased after that.

The non-catalytic curves for DEEA were plotted in Figure 3 and Figure 4. In this time period, amine absorption was recorded from fresh solvent to equilibrium solubility of 0.87 mol/mol, the rest of the data was not displayed because CO_2_ was reacting with H_2_O. It took 40 min for 1.0 mol/L and 45 min for 1.5 mol/L. With the aid of solid alkalis CaCO_3_, the time was reduced to 24 min (21 min less) for 1.0 mol/L and 30 min (15 min less) for 1.5 mol/L at optimized conditions. With the aid of MgCO_3_, the time was reduced to 24 min (21 min less) for 1.0 mol/L and 45 min for 1.5 mol/L at optimized conditions. The effect of MgCO_3_ was similar to that of CaCO_3_ at 1.0 mol/L, but not very helpful at 1.5 mol/L with 5 g. If bigger than 5 g MgCO_3_, the absorption curves were worse than the non-catalytic curves due to “agglomeration” [40]. Therefore, CaCO_3_ was a better solid for DEEA than MgCO_3_.

The non-catalytic curves for 1DMA-2P were plotted in Figure 5 and Figure 6. In this time period, amine absorption was recorded from fresh solvent to equilibrium solubility of 0.81 mol/mol. It took 35 min for 1.0 mol/L, 30 min for 1.5 mol/L. The bigger amine concentration was, the less time it would take, due to faster absorption rates [5] and the smaller cyclic capacity (0.81 mol/mol). With the aid of CaCO_3_, the time was reduced to 18 min for 1.0 mol/L, 27 min for 1.5 mol/L at optimized conditions. With the aid of MgCO_3_, the time was reduced to 17 min for 1.0 mol/L, 27 min for 1.5 mol/L at optimized conditions. The effect of CaCO_3_ was comparable to that of MgCO_3_. It was quite effective at 1.0 mol/L when the time was reduced by 18 min, and the time was reduced by only 3 min at 1.5 mol/L. 

The optimized amount of solid base chemicals is presented in Table 2. The orders were different under different amine concentrations. For DEEA, it was 7.5 g > 10 g > 5 g > 0 g for 1.0 mol/L and 7.5 g ≈ 5 g > 0 g > 10 g for 1.5 mol/L for CaCO_3_. The catalytic absorptions were better than the non-catalytic absorptions. For MgCO_3_, it was 10 g > 5 g > 7.5 g > 0 g for 1.0 mol/L, but 5 g > 0 g > 10 g > 7.5 g for 1.5 mol/L. We repeated the experiments of Figure 4A,B and Figure 6B at least twice. This poorer effect of CaCO_3_ and MgCO_3_ at large amount to 1.5 mol/L DEEA was probably due to the “agglomeration” [40] of solid chemicals, where the liquid covered the solid surface area and inhibited the catalysis. Such phenomena were also reported by other researchers with 0 g > 50 g CaCO_3_ to 4.0 mol/L BEA + AMP amine blend [40]. For 1DMA-2P, it was 7.5 g > 5 g > 10 g > 0 g for 1.0 mol/L and 10 g > 7.5 g ≈ 5 g > 0 g for 1.5 mol/L for CaCO_3_. For MgCO_3_, it was 10 g > 5 g > 7.5 g > 0 g for 1.0 mol/L and 5 g > 7.5 g ≈ 10 g ≈ 0 g for 1.5 mol/L. The larger amount of MgCO_3_ at 1.5 mol/L also resulted in agglomeration and made the catalysis comparable to non-catalytic absorption [40]. The removal of agglomeration awaits further investigation. 

Under both amine concentrations without agglomeration, the catalytic absorptions were better than the non-catalytic absorptions. For 1.0 mol/L, the absorption rates increased significantly with different amounts of solid chemicals. For 1.5 mol/L, the catalytic absorptions were better than non-catalytic cases at moderate ability. This difference was explained by Equations (13) and (14). At a dilute concentration of 1.0 mol/L the solid catalyst was helpful, for there are limited free R_3_N amines around H_2_CO_3_. However, at a higher concentration of 1.5 mol/L, there are more free R_3_N molecules in solution with higher pH value in solution, the reaction rate was increased with [R_3_N] and the solid chemical had only moderate improvements $r_{{\mathrm{CO}}_{2}}=k_{R_{3}N}\left[R_{3}N\right]\left[{\mathrm{CO}}_{2}\right]$.

### 4.4. The Effect of Solid Base to CO_2_ Absorption to Tertiary Amine DEEA and 1DMA-2P with Comparison to MEA and DEA

In addition to the periods of absorption profiles, the effect of CaCO_3_ and MgCO_3_ could also be evaluated by the initial absorption rates, which was an important parameter [33]. The initial absorption rates were shown in Figure 7 for non-catalytic absorption and optimized catalytic absorption. The effect of solid alkalis was not the more the better, and there was an optimized mass. According to Section 2.2 with Equation (13), the increased mass of solid alkalis helped the proton transfer from H_2_CO_3_ to solid surface at start. However, after the optimized mass from Equation (14), the excess amount of solid base inhibited the proton transfer from catalyst to R_3_N, and reduced the overall absorption rates. 

At optimized conditions, both rates increased significantly. For DEEA, the initial absorption rate was 0.74 × 10^−2^ mol CO_2_/min for 1.0 mol/L without catalysts, and it increased to 238% and 247% with the aid of CaCO_3_ and MgCO_3_. The initial absorption rate increased to 1.39 × 10^−2^ mol CO_2_/min for 1.5 mol/L, but increased to only 122% and 135% with CaCO_3_ and MgCO_3_. For 1DMA-2P, the initial absorption rate was 1.07 × 10^−2^ mol CO_2_/min for 1.0 mol/L without catalysts, and it increased to 153% and 150% with CaCO_3_ and MgCO_3_. The initial absorption rate increased to 1.24 × 10^−2^ mol CO_2_/min for 1.5 mol/L and increased to 165% and 149% with CaCO_3_ and MgCO_3_.

Compared with other studies of MEA and DEA, the absolute value of initial absorption rates of R_3_N was smaller than MEA and DEA, because of the lower absorption rates and smaller second order rate constant k_2_ [27]. With the aid of solid bases, the initial rates properly increased. The effect of solid chemicals was stronger at 1.0 mol/L and turned moderate at 1.5 mol/L for DEEA. For 1DMA-2P, the increase of initial absorption rates was similar at the range of 150–165% for 1.0 mol/L and 1.5 mol/L. The overall absorption periods were reduced by about 46–48% at 1.0 mol/L for DEEA and 1DMA-2P, but were reduced by only about 33% and 10% for 1.5 mol/L DEEA and 1DMA-2P.

Such a difference was due to the different reaction mechanisms of different reactions. For CO_2_ reaction with tertiary amine, it was the based catalyzed hydration mechanism. The R_3_N do not react with CO_2_ directly, but accept protons from [H_2_CO_3_]. The stronger basicity of the solvents led to stronger proton affinity, and caused better CO_2_ absorption. On the basis of kinetic studies, the second order rate constant (k_2_) was related to pKa of the tertiary amine. [5] The increased amine concentration led to a bigger pH value of the solution and higher absorption rates. The solid base chemicals could not directly affect the [OH^−^] or pH value in solvent, and provided only moderate enhancement of CO_2_ absorption rates at higher concentrations. In short, the solid alkalis were effective for tertiary amines, but the effects were not as good as for primary and secondary amines. They were more effective at a dilute concentration. 

However, for MEA and DEA, the CO_2_ reaction is driven by Zwitterion mechanism with the carbamate formation as products [28]. Solid alkalis might enhance the mass transfer or reduce the activation energy (Ea) of the reaction process and facilitate N-C bond formation of CO_2_-amine. The solid surface area contains abundant active sites which facilitate CO_2_ absorption in another reaction pathway [19,20].

### 4.5. The CO_2_ Desorption Profiles with Heat Duty Analyses

The CO_2_ desorption profiles were plotted in Figure 8 and Figure 9 for 1~2 mol/L DEEA and 1DMA-2P solvents. This range of amine concentration was suitable for industrial application, for the 0–2 mol/L MDEA were usually blended with 5M MEA to prepare MEA-R_3_N solvents, usually 5 + 0.5, 5 + 1, 5 + 1.25, 5 + 1.5 M [6,7,9]. The operation condition of MEA-R_3_N was also 0.20~0.60 mol/mol. Moreover, for 1DMA-2P, the solubility was low. Small amounts of crystal were observed in 2.0 mol/L solvent at 293 K, but it was soluble in water at 363 K.

From the CO_2_ desorption curves, there were some clues. For DEEA, the catalytic desorption was better than the non-catalytic one, with the order of H-ZSM-5 > γ-Al_2_O_3_ > non-catalyst at 5.0 g, 7.5 g and 10 g through 1.0~2.0 mol/L. For 1DMA-2P, the effects of H-ZSM-5 ≈ γ-Al_2_O_3_ > non-catalyst at 5.0 and 7.5 g, but H-ZSM-5 > γ-Al_2_O_3_ > non-catalyst at 10 g at optimized amount of H-ZSM-5. This effect was reasonable because the CO_2_ loading decreased from 0.80 to 0.30 mol/mol at the first 30 min. There were abundant bicarbonates [HCO_3_^−^], and more [R_3_NH^+^] from ion speciation plot, which made the CO_2_ production comparable with γ-Al_2_O_3_ and HZSM-5. Both solid acids could enhance the proton transfer process. However, at 10 g H-ZSM-5 of the series, an excess amount of H-ZSM-5 provided excessive protons into the solvents, and then fully reacted with [HCO_3_^−^] and produced more CO_2_ to reduce heat duty. 

The heat duty for the first 30 min was calculated in Figure 10 and Figure 11. The heat duty was mostly determined by the CO_2_ production (nCO_2_) as the heat inputs were similar for the first 30 min. During that period, most of the CO_2_ desorption process was completed as the CO_2_ loading decreased from 0.80 to 0.30 mol/mol. The CO_2_ desorption curves did not shift significantly after loading <0.30 mol/mol, since most [HCO_3_^−^] was exhausted. 

Based on Table 3, it was discovered that the heat duty was properly reduced. For γ-Al_2_O_3_, the order was 7.5 g > 10 ≈ 5 g > 0 g, and for H-ZSM-5, the order was 10 g > 7.5 g > 5 g > 0 g. At optimized catalytic conditions, the heat duty was reduced by about 83-98 % under different conditions. For DEEA, the reduction of heat duty followed the order of 1.5 > 1.0 > 2.0 mol/L. For 1DMA-2P, the reduction of heat duty followed the order of 1.5 > 2.0 > 1.0 mol/L. Hence, the amine concentration was preferred at 1.0–1.5 mol/L for DEEA and at 1.5–2.0 mol/L for 1DMA-2P with catalysts. 

### 4.6. The Effect of Solid Acid to Tertiary Amine DEEA and 1DMA-2P and Compared with MEA and DEA

The effect of solid acids to 1DMA-2P and DEEA was shown in Table 3. Briefly, the effects of H-ZSM-5 and γ-Al_2_O_3_ were different due to different reaction schemes and mechanisms. For tertiary amine, HZSM-5 was better than γ-Al_2_O_3_ for DEEA within the scope of 1–2 mol/L and 0–10 g. For 1DMA-2P, H-ZSM-5 was comparable to γ-Al_2_O_3_ with inadequate catalysis (5 g and 7.5 g), and better than γ-Al_2_O_3_ at 10 g. The increased mass of H-ZSM-5 led to better desorption performance, with the order of: 10 g > 7.5 g > 5 g > 0 g. 

This could be explained by the reactions (18) and (19) in Section 2.3. The Equation (18) reflected the fact that CO_2_ desorption rates were determined by mass of H-ZSM-5. Since H-ZSM-5 was a proton donor that reacted with bicarbonate right away, more H-ZSM-5 led to better desorption performance. After loading < 0.25 mol/mol, the effect of H-ZSM-5 was almost the same despite different masses because [HCO_3_^−^] was exhausted from ion speciation plot [17].

On the other hand, the effect of γ-Al_2_O_3_ was complicated. In some cases, 10 g was the best, and in other cases, 7.5 g was the best, which was even better than 10 g and 5 g. This trend was quite different from MEA [21] and DEA [26]. In short, for single catalysts, the increased mass of solid acid led to better desorption performance for MEA [21] and DEA [26]. From Appendix A [26] it’s concluded that the order was H-ZSM-5 > blended catalyst (γ-Al_2_O_3_/H-ZSM-5) > γ-Al_2_O_3_ at 0.50–0.30 mol/mol for MEA, blended catalyst (γ-Al_2_O_3_/H-ZSM-5) > γ-Al_2_O_3_ > H-ZSM-5 at 0.30–0.15 mol/mol for MEA [21]. For DEA, HZSM-5 was better than γ-Al_2_O_3_ in both rich and lean regions [26]. However, excessive amounts of γ-Al_2_O_3_ might reduce the catalysis of tertiary amines from optimum dose, while the effect was still better than that of non-catalyst. 

The reason was based on reactions (15)–(17), for the role of γ-Al_2_O_3_ was twofold. It had to accept [H^+^] from [R_3_NH^+^] first because it contained no proton, and then released [H^+^] to [HCO_3_^−^]. The release of CO_2_ was determined by Equation (17), and it was affected by [HAlO_2_]. From Equation (16), HAlO_2_ was generated from proton release from [R_3_NH^+^] to [AlO_2_^−^]. From Equation (15), the concentration of [AlO_2_^−^] increased with increased amounts of γ-Al_2_O_3_. The increased [AlO_2_^−^] enhanced the acceptance of [H^+^] in Equation (16), but the extra [AlO_2_^−^] might inhibit the proton transfer to [HCO_3_^−^] in Equation (17) and then affected the CO_2_ desorption rates. The increased mass of γ-Al_2_O_3_ enhanced desorption firstly, and then reduce desorption after the optimum, for the excessive amounts of [AlO_2_^−^] might inhibit the proton transfer from [HAlO_2_] to [HCO_3_^−^]. Similar to solid base (CaCO_3_ and MgCO_3_) to CO_2_ absorption, there was also an optimized dose of γ-Al_2_O_3_, and inadequate or excessive amounts of it would reduce the effects of CO_2_ desorption.

## 5. Conclusions

(1). The CO_2_ equilibrium solubility and pKa of DEEA and 1DMA-2P were comparable to published data, and the scope was expanded to 293 K from the previous 298–313 K.

(2). The existence of CaCO_3_ and MgCO_3_ as solid alkalis accelerated the CO_2_-R_3_N absorption, via the “base-catalyzed mechanism”. The effect of solid alkaline was indirect, it facilitated proton transfer from H_2_CO_3_ firstly, and the proton was then transferred to stronger base R_3_N. The increased mass of solid base boosted proton transfer, but the excess amount might inhibit the proton transfer from solid to R_3_N. Therefore, there was an optimized dose of CaCO_3_ and MgCO_3_ as catalysts. For solid alkalis, their effects were significant at 1.0 mol/L, but moderate at 1.5 mol/L because the increase of the amine concentration resulted in the increase of absorption rates and the increase of pH value. High amine concentration provided more free molecules into the solution to enhance the proton transfer of H_2_CO_3_.

(3). The solid acids could enhance the CO_2_ desorption and reduce the heat duties of both tertiary amines. The effect of catalytic desorption was better than that of non-catalytic ones. For DEEA, it was H-ZSM-5 > γ-Al_2_O_3_ > non-catalyst. For 1DMA-2P, it was H-ZSM-5 ≈ γ-Al_2_O_3_ > non-catalyst with inadequate catalysts, but H-ZSM-5 > γ-Al_2_O_3_ > non-catalyst at optimized performance. 

(4). The effect of Bronsted acid/proton donor H-ZSM-5 to CO_2_ desorption was straightforward, that is, the more the better as it reacts with bicarbonate directly. The effect of Lewis acid such as γ-Al_2_O_3_ to CO_2_ desorption was indirect. The increased mass of γ-Al_2_O_3_ resulted in increased [AlO_2_^−^], which could boost proton transfer of [R_3_NH^+^] to generate [HAlO_2_]. However, the excess amount of [AlO_2_^−^] might inhibit the proton transfer of [HAlO_2_] to [HCO_3_^−^] and release CO_2_. Therefore, there was surely an optimized dose of γ-Al_2_O_3_, which was not the more the better. 

## Figures and Tables

**Figure 1 molecules-24-01009-f001:**
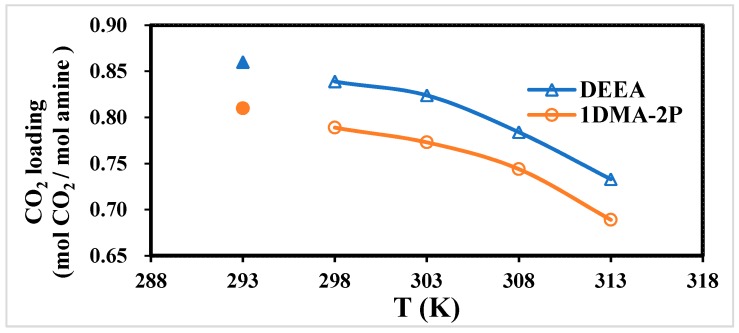
The critical point of CO_2_ absorption curves at 1 atm and 293 K, and CO_2_ equilibrium solubility of DEEA and 1DMA-2P at pressure of 1 atm and 298–313 K [5].

**Figure 2 molecules-24-01009-f002:**
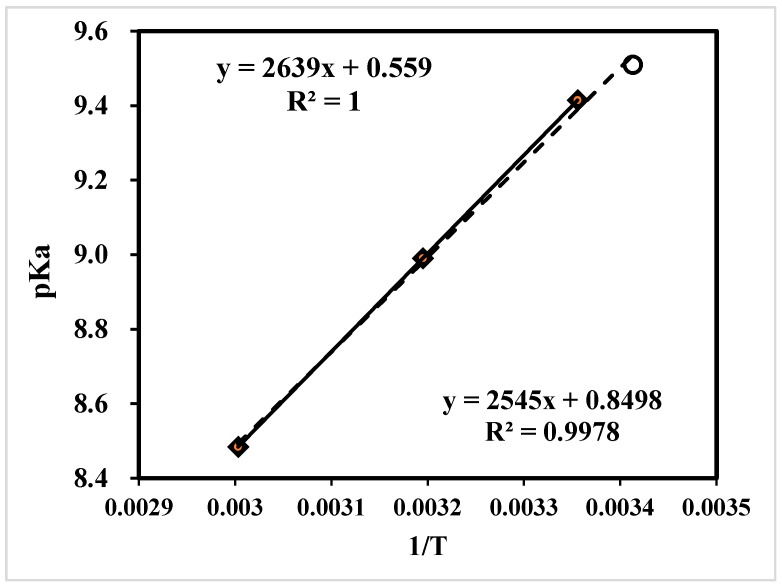
The pKa of 1DMA-2P at 293–333 K and 1 atm ^a^. ^a^ The equation above is for the pKa at 298–333 K and 1 atm from literature, [15] and the equation below is the calibration curve of 4 points at the range of 293–333 K.

**Figure 3 molecules-24-01009-f003:**
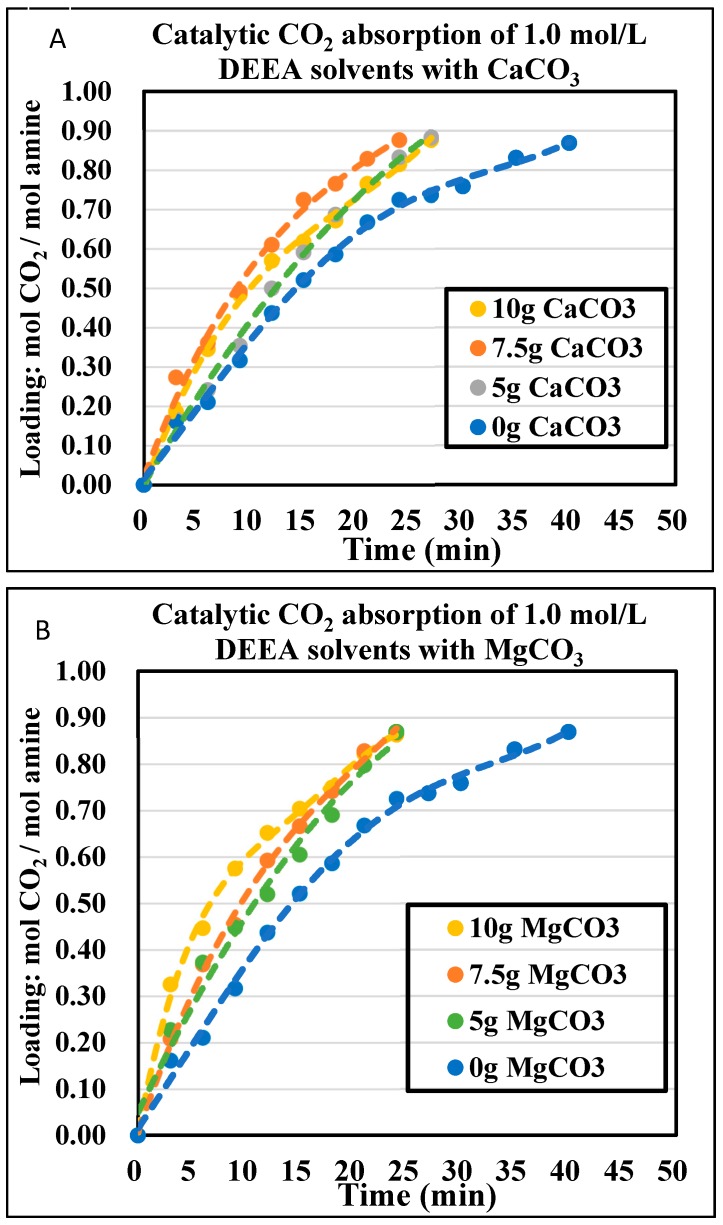
Catalytic CO_2_ absorption curves of 1.0 mol/L DEEA solvents at 293 K and 1 atm. (**A**) CaCO_3_ 0–10 g; (**B**) MgCO_3_ 0–10 g.

**Figure 4 molecules-24-01009-f004:**
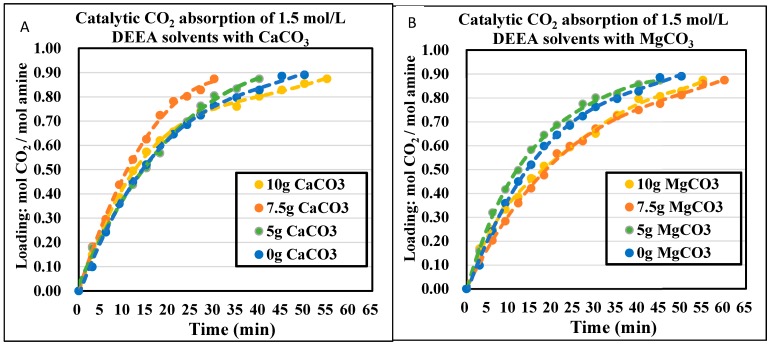
Catalytic CO_2_ absorption curves of 1.5 mol/L DEEA solvents at 293 K and 1 atm. (**A**) CaCO_3_ 0–10 g (**B**) MgCO_3_ 0–10 g.

**Figure 5 molecules-24-01009-f005:**
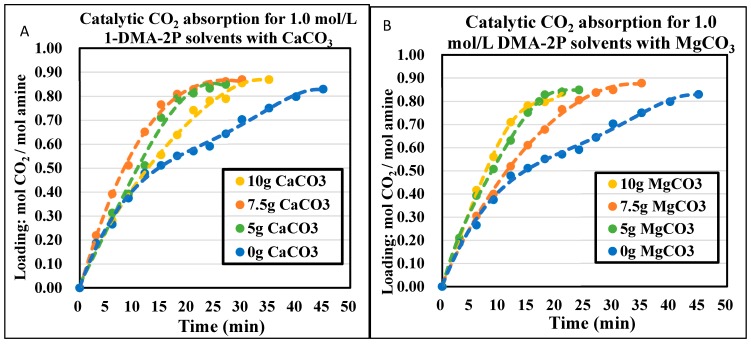
Catalytic CO_2_ absorption curves of 1.0 mol/L 1DMA-2P solvents at 293 K and 1 atm. (**A**). CaCO_3_ 0–10 g. (**B**). MgCO_3_ 0–10 g.

**Figure 6 molecules-24-01009-f006:**
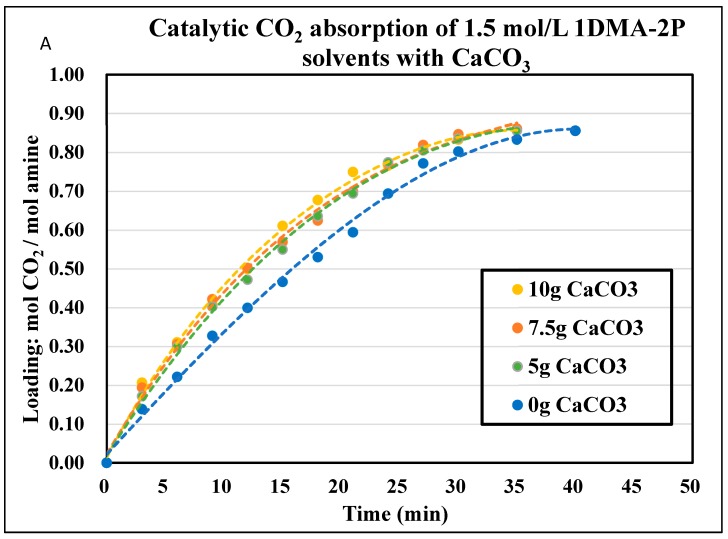
Catalytic CO_2_ absorption curves of 1.5 mol/L 1DMA-2P solvents at 293 K and 1 atm. (**A**) CaCO_3_ 0–10 g. (**B**) MgCO_3_ 0–10 g.

**Figure 7 molecules-24-01009-f007:**
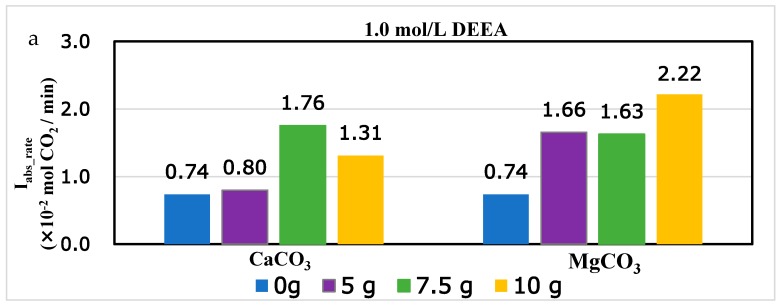
Initial Absorption rates for DEEA and 1DMA-2P with optimized amount of CaCO_3_ and MgCO_3_, from 1.0–1.5 mol/L at 293 K and 1 atm. (**a**) 1.0 mol/L DEEA, (**b**) 1.5 mol/L DEEA, (**c**) 1.0 mol/L 1DMA-2P, (**d**) 1.5 mol/L 1DMA-2P.

**Figure 8 molecules-24-01009-f008:**
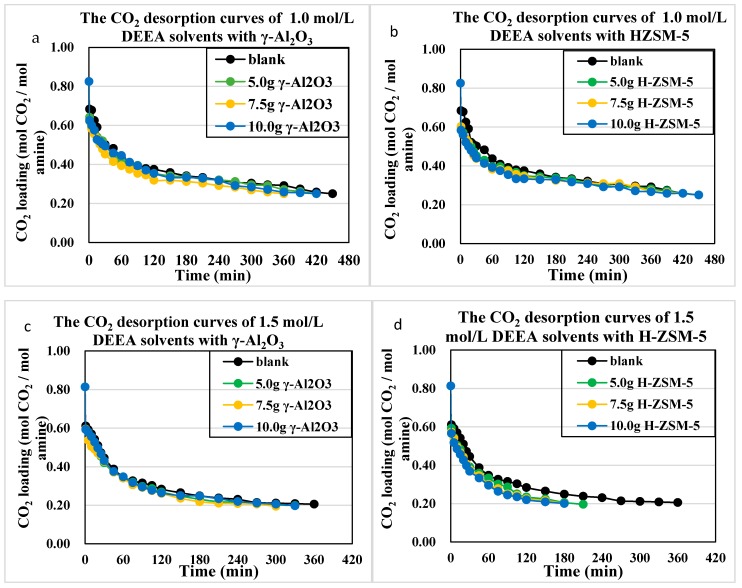
Catalytic CO_2_ desorption curves of 1.0–2.0 mol/L DEEA solvents at 363 K and 1 atm, with 0–10 g γ-Al_2_O_3_ and H-ZSM-5. (**a**,**b**) 1.0 mol/L with γ-Al_2_O_3_ and HZSM-5; (**c**,**d**) 1.5 mol/L with γ-Al_2_O_3_ and HZSM-5; (**e**,**f**) 2.0 mol/L with γ-Al_2_O_3_ and HZSM-5.

**Figure 9 molecules-24-01009-f009:**
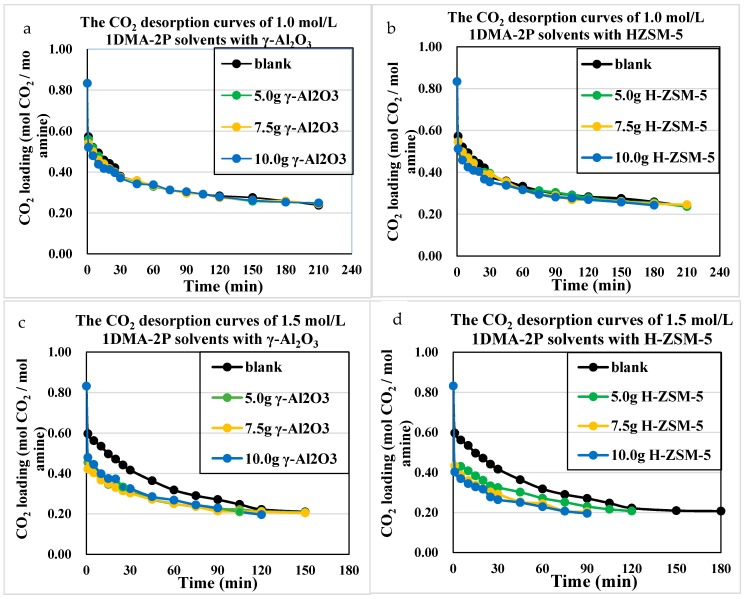
Catalytic CO_2_ desorption curves of 1.0–2.0 mol/L 1DMA-2P solvents at 363 K and 1 atm, with γ-Al_2_O_3_ and H-ZSM-5. (**a**,**b**) 1.0 mol/L with γ-Al_2_O_3_ and HZSM-5; (**c**,**d**) 1.5 mol/L with γ-Al_2_O_3_ and HZSM-5; (**e**,**f**) 2.0 mol/L with γ-Al_2_O_3_ and HZSM-5.

**Figure 10 molecules-24-01009-f010:**
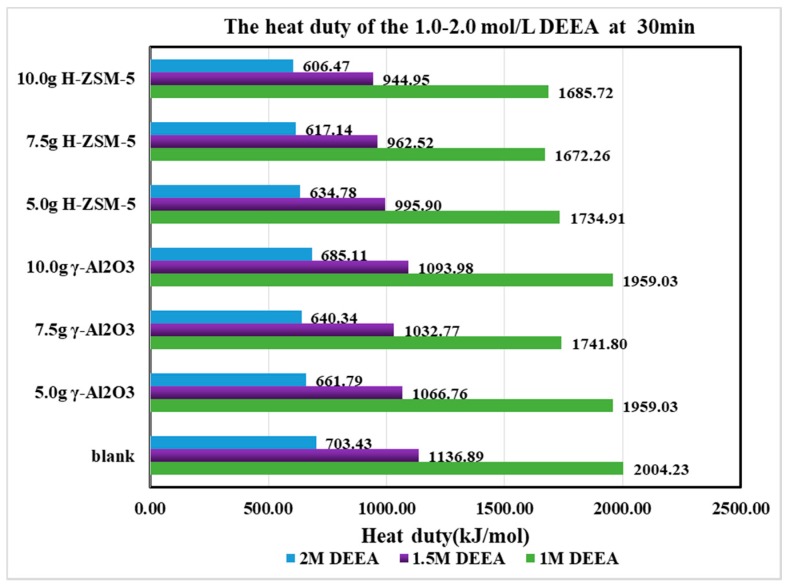
The heat duty of the DEEA at first 30 min from 1.0 to 2.0 mol/L at 363 K and 1 atm.

**Figure 11 molecules-24-01009-f011:**
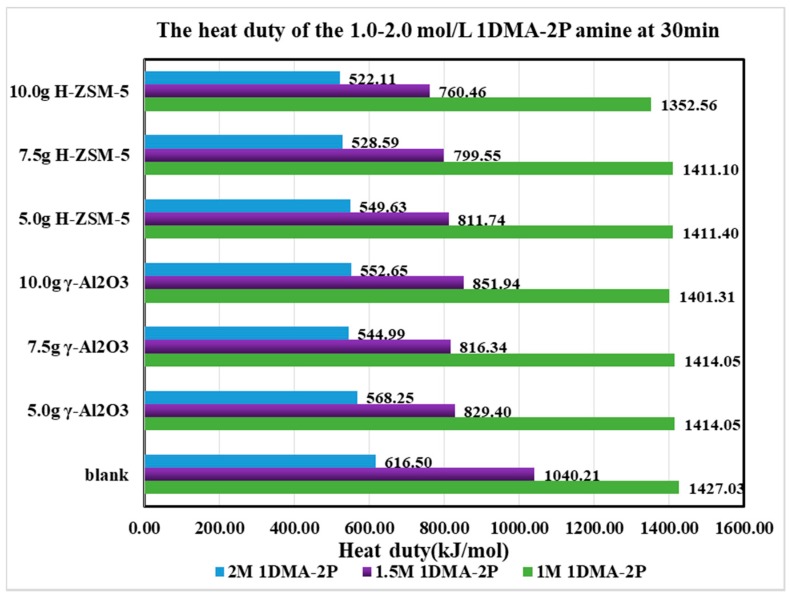
The heat duty of 1DMA-2P at first 30 min from 1.0 to 2.0 mol/L at 363 K and 1 atm.

**Table 1 molecules-24-01009-t001:** pKa of investigates amines at 293 K and 1 atm.

Amine	Predicted pKa	Reference	Measured This Work
DEEA	9.60 (298 K) [5]	9.73 (298 K) [11] 9.82 (293 K) [11]	9.82 (293 K)
1DMA-2P	9.20 (298 K) [5]	9.67 (301 K) [17] 9.41 (298 K) [15]	9.51 (293 K)

**Table 2 molecules-24-01009-t002:** The optimized mass of CaCO_3_ and MgCO_3_ for different amine solvents at 293 K and 1 atm.

Amine Solvents	CaCO_3_ (g)	MgCO_3_ (g)
1.0 M DEEA	7.5	10
1.5 M DEEA	7.5	5
1.0 M 1DMA-2P	7.5	10
1.5 M 1DMA-2P	10	5

**Table 3 molecules-24-01009-t003:** The relative heat duty (%) of different amine solvents under optimized catalysis at 363 K and 1 atm.

Amine Solvents	Optimized Catalysts	Heat Duty (kJ/mol CO_2_)		
		Optimized Catalysis	Non-Catalyst	Ratio (%)
DEEA 1.0 mol/L	7.5 g γ-Al_2_O_3_	1741.8	2004.2	86.91%
	10 g HZSM-5	1685.7	2004.2	84.11%
1.5 mol/L	7.5 g γ-Al_2_O_3_	1032.8	1136.9	90.84%
	10 g HZSM-5	945.0	1136.9	83.12%
2.0 mol/L	7.5 g γ-Al_2_O_3_	640.3	703.4	91.03%
	10 g HZSM-5	606.5	703.4	86.22%
1DMA-2P 1.0 mol/L	10 g γ-Al_2_O_3_	1401.3	1427.0	98.20%
	10 g HZSM-5	1352.6	1427.0	94.78%
1.5 mol/L	7.5 g γ-Al_2_O_3_	816.3	1040.2	78.48%
	10 g HZSM-5	760.5	1040.2	73.11%
2.0 mol/L	10 g γ-Al_2_O_3_	545.0	616.5	88.40%
	10 g HZSM-5	522.1	616.5	84.69%

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
