# Peer review of "Study of Catalytic CO2 Absorption and Desorption with Tertiary Amine DEEA and 1DMA-2P with the Aid of Solid Acid and Solid Alkaline Chemicals"

_molecules, 2019, doi:10.3390/molecules24061009_

Round 1
Reviewer 1 Report
The authors of this manuscript studied the effect of Acid/base catalysts on the co2 solubility for Absorption/desoprtion with auqeous amine Solutions.
The work is interesting, the results are new. However, I have the Feeling that the work suffers from several drawback such as precision, data reduction, and missing or wrong conclusions from the data.
Thus, my recommendation is between reject and Major Revision (if the authors are willing to present new data).
In Detail:
All the data the quthors present are very strong dependent on pressure. THe pressure is of central importance. Pressure has to be given not as a hidden Information. With which accuracy could the pressure be determined?
Fig. 1 Authors Claim that their data has solubility of About 82 and 86 x% for DMA and DEEA, respectively. However, this does not fit to the diagrams 4 and 6. If the data in figs 4+6 are correlated until Equilibrium Steady state,values of 87x% and 95x% for DMA and DEEA might be possible. This is an inconsistency, and the authors should add new data to figs. 4+6 with Experiments at longer time (e.g. 50 minutes).
How was the concentraiton of Equilibrium solubility measured (the technique?)
The curves in fig. 6 A +B for Zero CO3 salt look very strange. What has happened there? Is this reproducible? How many time were the measurements done?
I do not understand the results of fig. 7. sometimes the kinetic constants increase with increasing catalyst concentration, and at a certain Point it decreases again. What is the Explanation? It is not discussed at all.
if the catalyst changes pH, then this might indeed influence the solubility at Equilibrium. Authors should give together with the solubility data also the pH, in order that one can separate between pH effects and catalyst Addition. Maybe this also woul explain fig. 7.
authors have meaured in fig. 4+6 only each 2-3 minutes. Especially in the beginning of the kinetic Experiments, data is very sensitive and Errors have big influence on the kinetic constants. AUthors used eq. 23. But they should give the time (t=0 until t=???) range at which they used eq. 23. Otherwise almoste every result can be "produced" by manipulating the time range.
Authors should be more concise with Parameters and expressions. what is k2, kR3N? pH meter does NOT measure concentration of H+, rather ist activity.what means "under VLE model". nobody can understand.
authors conclude that catalysts help only at rather dilute conditions of amine. However, technically, dilute Solutions are not relevant at all. Rather concentrated amine Solutions are used for the process in Industry.
Author Response
Dear Reviewer:
We put a a point-by-point response to the reviewer’s comments in the attach file.

Reviewer 2 Report
The manuscript focuses on the absorption/desorption behavior of CO2 using two tertiary amines with aid of solid alkaline chemicals. The manuscript is hard to follow, the novelty is not clear since both of the studied tertiary amines have been previously introduced and investigated. The way also the figures are presented, it does not show significant performance differences between different scenarios. For example, the figure illustrated on page 12, does not show a difference between the "blank" and other scenarios.
Author Response
Dear Reviewer 2:
I provided a point-by-point response to the reviewer’s comments in attach file.

Reviewer 3 Report
This paper is a study of CO2 capture with solid alkaline as catalysts for absorption and solid acid for desorption. Effects of amine concentration and catalyst amount on the catalytic performance were investigated. However, substantial improvements need to be done before it can be reconsidered for the publication in Molecules. Followings are my major concerns:
1. An updated and complete literature review should be conducted. Some related articles should be cited, such as ACS Sustain Chem. Eng. 5, 5862–5868 (2017) and Nat. Communi. 9, 2672, (2018).
2. Reference in table one should be given.
3. The quality of Figure3, 4, 6 and 8 should be improved.
4. How can you prove that it is the catalytic effect, not just transportation effect? The addition of solid particles to a stirred reactor will result in enhanced gas-liquid mixing and increased rates of mass transfer and thus increased adsorption. Also, it is possible that the high surface area solids can catalyze the nucleation of CO2 bubbles that provide more mass transfer surface for the desorption process. The authors should rule out the possible significant effect of mass transfer causing on CO2 absorption and desorption.
5. Solvents with 10 g CaCO3 and MgCO3 show much better performance than that without catalysts, when the concentration of DEEA is 1.0 mol/L. But, solvents with 10 g CaCO3 and MgCO3 show even lower performance than that without catalysts, when the concentration of DEEA is 1.5 mol/L. How do you explain why the slight increase of DEEA concentration from 1 to 1.5 mol/L would greatly affect the performance of solvents with catalyst?
6. The definition of C and V in eqn(23) should be given. Which are the data used for calculation, first 5, 10 min or others?
7. Data of subfigures of Figure 10 can be potted as one figure. The same issue needed to be addressed for Figure 11.
Author Response
Dear Reviewer 3:
We completed a point-by-point response to the reviewer’s comments.

Round 2
Reviewer 1 Report
1. In my original review question 1) I asked to give the pressure in each table and diagram as figure or table captions. This is very important, and still needs to be done (e.g fig 1).
2. The same is valid for the references to experimental data. If data is shown that comes from literature, this has to be cited in the figure captions (e.g. fig1)
3. Solubility is an equilibrium property. IF the authors define that differently, they must NOT call this value solubility. Solubility will be at tàinf. It does not make any sense at all to define solubility with the requirement that all amine is present as ionic species. A procedure to calculate solubility with slopes is inconsistent and forbidden from thermodynamic point of view. The final point is always the solubility in the studied system. This is especially important if other researchers start modeling the system.
Thus, the authors have clearly to define which CO2 concentration they measured, and that they did NOT study the equilibrium solubility. The differences might not be large, but they are still also not neglectable.
4. I am surprised that the introduction does not mention predictive models for CO2 solubilities in amine solutions (e.g. https://dx.doi.org/10.1016/j.fluid.2017.12.033 https://dx.doi.org/10.1016/j.fluid.2015.02.026)
Author Response
We provided a point-by-point response to the reviewer’s comments, and the main Changes are in Fig 1 and Section 4.1.
Please check the attach file.

Reviewer 2 Report
I think in the revised version of the manuscript, the novelty of the work has been clearly emphasized. So, I recommend the manuscript for publication.
Author Response
The reviewer recommend the manuscript for publication already.
There is no other comments.
Reviewer 3 Report
Thanks for the modification and clarification from the authors. The present version has addressed most of reviewer's concerns and the quality of the manuscript has been improved. However, minor revisions are still needed. The followings are my concerns:
1. Following is the link of of Nat. Communi. 9, 2672, (2018) mentioned in my previous comment: https://www.nature.com/articles/s41467-018-05145-0. Please add this one to the revised manuscript and make necessary change to current Ref. 24, removing “and Nat. Communi. 2018, 9, 2672.” from it.
2. “Base-catalyzed hydration mechanism” in line 22 should be “base-catalyzed hydration mechanism”
3. The quality, not just the resolution, of figures can be further improved. For example, some titles of Y axis overlap numbers of Y axis in Figure 7, which should be addressed. Also, for figure 8 and 9, the major units of X and Y axis can be increased to show less numbers on the axis. Moreover, the titles for each sub-figures in figure 8 and 9 are not necessary.
Author Response
We provided a point-by-point response to the reviewer’s comments.
Fig 7, 8-9 were modified with both contents and Captions.
The Reference were added as Ref 25, with content in Introduction.
